# Motor Imagery as a Key Factor for Healthy Ageing: A Review of New Insights and Techniques

**DOI:** 10.3390/brainsci12111492

**Published:** 2022-11-03

**Authors:** Noemi Passarello, Marianna Liparoti, Caterina Padulo, Pierpaolo Sorrentino, Fabio Alivernini, Beth Fairfield, Fabio Lucidi, Laura Mandolesi

**Affiliations:** 1Department of Humanities, Federico II University of Naples, 80138 Naples, Italy; 2Department of Social and Developmental Psychology, Faculty of Medicine and Psychology, “Sapienza” University of Rome, 00185 Rome, Italy; 3Department of Psychological, Health and Territorial Sciences, Gabriele d’Annunzio University of Chieti, 66100 Chieti, Italy; 4Institut de Neuroscience des Systemès, Aix-Marseille University, 13005 Marseille, France

**Keywords:** motor learning, healthy ageing, mental exercise, neuroimaging

## Abstract

Motor imagery (MI) describes a dynamic cognitive process where a movement is mentally simulated without taking place and holds potential as a means of stimulating motor learning and regaining motor skills. There is growing evidence that imagined and executed actions have common neural circuitry. Since MI counteracts cognitive and motor decline, a growing interest in MI-based mental exercise for older individuals has emerged. Here we review the last decade’s scientific literature on age-related changes in MI skills. Heterogeneity in the experimental protocols, as well as the use of populations with unrepresentative age, is making it challenging to draw unambiguous conclusions about MI skills preservation. Self-report and behavioural tasks have shown that some MI components are preserved, while others are impaired. Evidence from neuroimaging studies revealed that, during MI tasks, older individuals hyperactivate their sensorimotor and attentional networks. Some studies have argued that this represents a compensatory mechanism, others claim that this is a sign of cognitive decline. However, further studies are needed to establish whether MI could be used as a promotion factor to improve cognitive functioning and well-being in older people.

## 1. Introduction

Since life expectancy is increasing across the globe, it is important to identify factors that impact the quality of ageing. In addition to being largely viewed as a positive consequence of medical, social, and economic advancements over disease, global ageing also presents unique challenges and opportunities for our healthcare and social systems. World Health Organization (WHO) defines healthy ageing (HA) as developing and maintaining functional ability to enable well-being in old age. This definition of HA includes a state of complete physical, mental, and social well-being, not just the absence of diseases or infirmities.

Several studies have been conducted to identify factors contributing to physical, social, and psychological well-being, in order to promote healthy ageing (HA). Rowe and Kahn [1] defined HA as having a low risk of disease and disease-related disability, high cognitive and physical functional capacity, and being actively engaged in life. Despite this being one of the most conclusive definitions of HA, researchers still have not reached an agreement on how this concept is defined. Therefore, identifying factors that contribute to HA remains a difficult challenge for recent studies.

Atallah et al. [2] have found several potentially modifiable lifestyle factors that could influence the quality of ageing, such as smoking status, physical activity, and diet. Physical activity has been shown to affect brain plasticity, improving cognition and well-being [3,4,5,6]. Experimental and clinical studies have reported that physical activity induces structural and functional changes in the brain, determining enormous biological and psychological benefits [4,7,8]. Further, increasing physical activity and sport practice seems to prevent cognitive decline associated with ageing, to reduce the risk of developing dementia, and to prevent deterioration in executive function [9,10,11]. Hence, maintaining an “enriched lifestyle” until middle age has a positive effect on cognitive function [12]. Combined with many other experiences, physical activity provides a “reserve”-like advantage that maintains cognitive function in old age [13].

As physical activity plays a significant role in promoting HA, and neural networks involved in motor behaviour overlap with those that are involved in imagined motor behaviour, several researchers have questioned whether motor imagination processes, which are known to be more developed in active individuals [14], also contribute to a successful aging process.

The concept of motor imagery (MI) describes a dynamic cognitive process where a movement is mentally simulated without actually taking place [15]. A growing body of evidence indicates that imagined and executed actions share similar characteristics, particularly in terms of their temporal characteristics and the neural activity they entail [16]. These similarities have strengthened the interest in mental practice based on MI (or MI training), by which movements are repeatedly envisioned with the intention of improving their execution [17].

Sharing the same neural circuits of execution, MI can prove to be a useful intervention strategy in many conditions involving advanced age that do not allow for adapted physical activity. For example, to recover a motor ability [3,14]. Moreover, it is important to underline that any brain activity, such as in this case the imagination of motor act, increases the processes of brain plasticity and allows to face with age-related neurodegenerative diseases [3,4,8]. MI-based interventions could reduce the effects of age-related decline in motor function, which affects gait, balance, and coordination, and eventually preserve functional autonomy in the older [18].

However, to understand if MI-based training can be a useful tool for promoting older people’s psychophysical well-being, it is necessary to establish whether MI abilities are preserved during ageing. In this regard, some studies have examined how ageing affects MI and its facets, such as the vividness of motor representations (i.e., the ability to generate vivid images and sensations in the mind); the timing of MI (i.e., reproducing the duration of a movement during a mental simulation); the controllability of MI (i.e., manipulating a mental representation of a movement); and the accuracy of MI (i.e., the accuracy of motor representation) [19]. Methodological problems are reported in all these studies—owing to heterogeneity in experimental samples, measurements, tools used, and MI characteristics—which makes it necessary to present a comprehensive report of all scientific evidence. Studying the neural correlates underlying MI processes could also solve these issues and provide accurate information on the preservation of MI in healthy ageing.

We conducted this narrative review to gather information on motor imagery abilities in healthy ageing, examining whether MI and its components are preserved in older individuals, and what brain mechanisms support these competencies.

## 2. Method

Our narrative review was conducted by analysing scientific papers published between 2010 and 2022. We searched published papers on Pubmed, ScienceDirect, and PsychArticles databases, using “motor imagery”, “healthy ageing”, “healthy older” as keywords. Seventeen studies were selected. Among inclusion criteria there were: (a) experimental studies published in indexed journals; (b) studies regarding the role of motor imagination in healthy ageing, with a focus on its neural correlates and application to health-promoting training; (c) studies examining healthy older populations; (d) studies published in English; (e) studies available online. Exclusion criteria included: (a) studies not directly related to MI in healthy ageing; (b) studies on populations with cognitive impairment, head trauma, medical conditions such as stroke, younger populations, and populations under 64 years of age; (c) reviews, meta-analysis, and grey literature.

## 3. Motor Imagery in Healthy Ageing

To understand whether the use of MI in rehabilitation or health promotion training is appropriate, it must be established whether MI abilities are actually preserved during ageing. As pointed out by Schott [20], cognitive and sensorimotor functions show a linear decline during the old age. Considering how imagery supports various cognitive and motor functions, it stands to reason that less efficient imagery processes might contribute to older adults’ struggles with relearning daily living activities. There is, however, no consensus on this scientific evidence, since several studies have shown that MI abilities are instead preserved as we age [17,18,21].

The lack of agreement among the studies seems to depend on two main factors: (1) the assessment of different MI components; (2) the heterogeneity in methodologies used. In the following two sections, we will analyse both issues.

### 3.1. Motor Imagery Components

It is generally agreed that MI is a multidimensional construct. Thus, it is crucial to consider all of its different features when studying its age-related changes. To our knowledge, few studies have dealt with the investigation of different facets of MI and their changes during ageing. Moreover, most of these papers show conflicting results and heterogeneous methodologies. Later we will address the different methodology used in these studies, while in this section we will discuss the scientific evidence regarding the evolution of each MI component with age.

The term MI vividness is used to indicate the ability to generate vivid motor images and sensations in the mind [17]. Scientific literature is divided on the evolution of MI vividness with age. This is largely due to the different aspects of this MI component that are taken into account by these studies. When assessing MI vividness, individuals can mentally simulate movements from a first-person perspective (i.e., as if one is the actor of the action) or from a third-person perspective (i.e., as if one is a spectator of the action). When individuals imagine a motor action in first-person perspective, they mostly experience kinaesthetic sensations, as if they were actually performing the action; in third-person perspective, MI visual representations are the most significant, as if the individual were watching themselves or others perform the action [18]. While both modalities (kinaesthetic and visual) are investigated in studies on MI vividness, they are also not always differentiated from one another. Of all our selected studies, two reported that both young, middle-aged, and older adults showed good MI vividness skills. However, while in the young and middle-aged adults significantly higher visual than kinaesthetic abilities were found, this visual dominance was no longer observed in the older [21,22]. The latter result was not replicated by the other two studies, where older adults’ MI vividness skills seemed to be preserved. In both studies, their performance was equivalent to that of the young in either kinaesthetic or visual mode [17,23]. A final study showed that older adults (aged between 70 and 79) showed a general decline in MI vividness. Nevertheless, MI vividness skills were comparable between young participants and middle-aged or older (aged between 60 and 69), showing no differences between kinaesthetic and visual modalities. Interestingly, the decrease in physical activity, especially in the older adults, led to lowered kinaesthetic input, that compromised MI abilities [20].

MI timing component refers to the duration of a movement during a mental simulation. According to the literature, the amount of time required to imagine a motor action is expected to be close to the one needed to execute it [24]. Three of the studies that investigated age-related differences in MI timing, are in agreement in reporting a relative retention of this skill in healthy older [17,20,21] Yet, other findings suggest that while in young adults the temporal congruence of MI (i.e., the difference between the movement execution and imagination time) is not affected by movements’ constraints, in older adults this skill is not always maintained when they imagine constrained movements. In a study by [25], it was found that temporal congruence between imagined and executed movements sequence decreased with age. In this case, participants had to imagine and execute a series of fast and accurate arm movements between targets of decreasing sizes. This result was replicated in a similar study by [26], where, however, it was also shown that visual cues could improve MI performance in older, even if the motor sequence was composed by constrained movements. It is interesting to note that, again, physical activity could represent a protective factor for the maintenance of MI timing skills in the older. Several studies have already proven that MI skills are better developed in individuals who regularly engage in physical activity [14]. Evidence on active and sedentary older populations is needed to prove this beneficial role of physical activity in MI timing skills.

MI controllability refers to the ability to manipulate a mental representation of the movement [17]. Among our selected studies, only two assessed MI controllability. In both studies, a deterioration of this skill with the advancing of age was shown [17,20].

Lastly, MI accuracy was assessed by three studies. While two of these reported poorer performance in older [27,28], one reported no differences between young and older individuals [29]. The heterogeneity of the results concerning this MI component serves three possible explanations: (1) the studies used tasks that were too different in terms of their difficulty; (2) the studies examined young populations against highly older populations (over 70); (3) the majority of the tasks used in these studies failed to consider the possible impact of other cognitive factors, like planning, problem-solving, or depth perception.

Most of our selected studies have highlighted that the maintenance of specific cognitive abilities is crucial for the preservation of MI abilities in old age. Among these skills, working memory seems crucial to MI vividness and timing [17,20,21]; mental rotation processes and cognitive flexibility are associated with MI controllability [29]; planning and problem-solving skills appear fundamental for MI accuracy [28].

However, only one study by Schott [20] has actually evaluated the mediating effect of working memory on MI components during ageing. Specifically, findings from this study suggest that ageing in itself is not a potentially impairing factor for MI, rather it is the decline in cognitive functions, like working memory and attention, together with changes in activation patterns of several brain regions that can result in a loss of vividness and timing of MI.

In conclusion, it must be stressed how most of the studies analysed so far failed to provide a sample that was adequately representative of the healthy older population. Some studies used samples with individuals who were too old, others with those who were too young. It is essential to increase the number of studies that consider the ‘young-old’ population (aged from 60 to 69 years old), as it seems that in that age range, some MI components are nearly always preserved.

### 3.2. Heterogeneity in Methodology

Self-report questionnaires and behavioural tasks have provided most of the evidence on MI in healthy ageing. Our research showed that in the last decade, nine studies have combined both of these methods to study MI in older populations.

Four of our selected studies use the Kinaesthetic and Visual Imagery Questionnaire (KVIQ) [30]. In addition to being intended for people with physical disabilities, this questionnaire has also been adapted to suit older individuals. In a study by Saimpont et al. [17], MI abilities were assessed in both young and older participants using KVIQ short versions. There were 5 items in this version, corresponding to 5 basic movements: one boot movement and four proximal and distal movements of the upper- and lower-limbs. During the test, participants were asked to first execute and then imagine—from a first-person perspective—the different movements. Each imagined movement was then rated on a 5-point Likert scale based on the clarity of the images (visual modality) or the intensity of the sensations produced (kinaesthetic modality). This study demonstrated that MI skills (specifically MI vividness) were similar among young and older. The same results were found by Heremans et al. [23] using the extended version of the questionnaire. The extended version of the KVIQ was also used by Malouin et al. [21]. This version included 10 basic movements and was used to assess the vividness of kinaesthetic and visual first-person MI in young (men age = 26.0) middle-aged (mean age = 53.6) and older (mean age = 67.6) participants. Contrary to the above, both this study and a similar one by Saimpont et al. [22] reported lack of visual dominance during first-person MI in older. Among our selected studies, only one used the Movement Imagery Questionnaire Revised (MIQ) [31]. Schott [20] used this tool to assess age-related differences in MI. In the MIQ, eight tasks were used to measure visual and kinaesthetic mental practice of movements. A variety of simple upper-extremity, lower-extremity, and whole-body movements are included in the questionnaire. For each movement, participants were asked to first execute the movement and then—after returning to the starting position- they had to imagine it. Finally, participants rated the difficulty experienced in imagining the movements on a 7-point Likert scale, ranging from 1 (‘‘very hard to see = feel’’) to 7 (‘‘very easy to see = feel’’). Study results showed that young adults’ imagery abilities were superior to those of older adults aged 70 and older, but not those aged 60 to 69.

Besides vividness, MI’s other components (specifically timing, controllability, and accuracy) were mostly studied through behavioural tasks.

Four of our selected studies used behavioural task to assess MI timing in young and older individuals. Both Malouin et al. [21] and Schott [20] used the Timed Up and Go test (TUG) [21]. In this test, participants were asked to first imagine and then execute five simple tasks—rising from a chair, walking 3 m forward, turning around, walking back to the chair, and sitting down. Imagination condition was presented first to minimize the possibility that participants were influenced by the duration of the real movement execution. Each performance was timed by the experimenter, and the imagination and execution times were compared. Moreover, both Saimpont et al. [17] and Schott [20] used mental chronometry tests to evaluate MI timing skills in their participants. In the first study, young and older were asked to physically and mentally perform a sequence of foot movements. From a starting position, in which their feet were resting on the floor on two black crosses, participants had to move their feet at a natural, self-selected speed, to hit different targets, in the following order: (1) left foot to left target; (2) right foot to middle target; (3) left foot to middle target; and (4) right foot to right target. After the execution trail, they had to imagine the complete sequence of movements. Both trials were recorded and evaluated. In the second study, young and older were tested through the Walking Test [32]. This time, participants were requested to execute and to mentally simulate walking movements towards a cone target, placed at eight different distances. The duration of executed and imagined movements was recorded by the experimenter. Finally, Heremans et al. [26] used a different task to assess MI timing. They asked young and older adults to transport 20 blocks physically and mentally with their hands, as fast as possible, from one side of a box to the other. The imagination phase was performed under three conditions: (a) with visual cues (i.e., subjects were allowed to see the box and the blocks during the trial); (b) with auditory cues (i.e., no vision of the box and blocks was provided, but a metronome paced the movements of the subjects during the trial); (c) without visual or auditory cues. Although they used heterogeneous tools, all these studies reported that older MI timing skills were comparable to those of the young individuals.

Similar to the above studies that analysed age-related differences in MI controllability have also shown an overall agreement in their findings, despite the heterogeneity of the methods used. Of all our selected studies, two used behavioural tasks to assess MI controllability. Both studies reported a lack of MI controllability in older individuals. Saimpont et al. [17] used a finger–thumb opposition task, during which participants had to repeatedly execute and imagine—from a first-person perspective—an auditory paced (1 Hz) finger–thumb opposition sequence with their dominant hand. The sequence consisted of touching fingers 2 (index), 4, 3, 5 and was repeated until the pacing stopped. Participants were required to start the finger movements at the onset of the first low-pitched sound, and to continue the sequence until the last sound. After each trial, they indicated which finger they had moved/imagined with the last sound. The response could be either correct (expected finger) or wrong (any of the three other fingers). The Controllability of Motor Imagery (CMI) test was instead used by Schott et al. [20]. This test evaluates the ability to supplement, transform, and reconstruct one’s internally visualized body schema in response to verbal instructions. In this study, two conditions were performed: a recognition test (controllability of body schema) and a regeneration test (ability to transform visual imagery). Participants had to perform 10 sets of trials with six consecutive instructions. In each trial, they were asked to imagine that they were moving their body parts according to the verbal instructions. On the regeneration test, the participants had to actually execute the final position immediately after the instructions were completed. On the recognition test, they were required to select among five pictures the one that fit the imagery they had. Their performance was scored by the experimenter.

As previously mentioned, among MI components, vividness and accuracy seem to be the ones with most disparate results between studies. Likewise, behavioural tasks used to assess it are also heterogeneous. Three of our selected studies used different tasks to evaluate MI accuracy. Devlin and Wilson [29] required young and older subjects to solve a hand-laterality task, in which pictures of hands were presented in a back view and in six different orientations. All participants used their dominant hand to complete the tasks by pressing one of two appropriately labelled keys when deciding whether the stimulus was a left or right hand. Response times’ results indicated that both age groups performed well in the task, by mentally moving their upper limbs. On the other hand, Gabbard et al. [27] showed that older were less accurate than young participants, in a task where they had to estimate, using MI, whether randomly presented targets in peripersonal (within actual reach) and extrapersonal (beyond reach) space were within or out of reach of their dominant upper limb while seated. Lastly, Saimpont et al. [28] used a puzzle task in which young and older individuals were required to put in order six images, depicting the main movements necessary to get up from the floor. Results showed that only 68% of the older subjects succeeded in the task.

As can be seen from the analysis of these studies, heterogeneity in the experimental protocols is making it challenging to draw unambiguous conclusions about MI skills preservation in healthy older people. Self-report tests and behavioural tasks, while essential to MI research, have failed to provide a comprehensive picture of MI evolution during ageing. Recent innovations in neuroimaging techniques could help overcome these limitations. Our next section will focus on analysing studies that have used these techniques.

## 4. Neural Correlates of Motor Imagery in Healthy Ageing

A large body of literature has documented that changes in cognitive function associated with normal ageing are accompanied by age-related changes in brain activity [33]. As for motor functions, neuroimaging studies have demonstrated that older individuals’ brain activation patterns differ when executing both simple and complex movements [34]. However, fewer neuroimaging studies on MI age-related changes are available. Our literature search yielded only seven neuroimaging reports on MI in healthy older, in the last decade.

Motor Imagery-related neuroimaging studies are based on the assumption that motor imagery and motor execution depend on partially overlapping neural systems. Dorsal premotor cortex, supplementary motor area, ventral lateral premotor cortex, intraparietal sulcus cortex, and supramarginal gyrus all belong to this shared neural network [35].

Our next two sections will review neuroimaging studies that have analysed MI in older populations, taking into account their limitations and proposing innovative methods for overcoming them.

### 4.1. fMRI Studies on Motor Imagery in Healthy Older

Of our selected studies, five used functional magnetic resonance imaging (fMRI) techniques, in combination with behavioural tasks, to analyse MI skills in healthy older individuals.

It appears that most fMRI studies agree that executive control, sensory-motor, and visuospatial brain networks are hyperactive in healthy older, during MI tasks. Zapparoli et al. [35] have used fMRI in combination with a chronometric test to assess MI timing skills in both young and older participants. They found significant neurofunctional differences between young and older subjects during the MI task. Specifically, they found an hyperactivation of older participants’ brains, primarily in the occipito-temporo-parietal areas. This evidence is supported by those studies that have linked parietal and occipital brain activity to visual imagery [36], mental image generation [37], spatial mental imagery and navigation [38]. In older, there was also a significant correlation between MI timing and the occipito-parietal hyperactivity. Researchers have replicated these results using a hand-laterality task to assess MI accuracy. In this case, occipito-parietal hyperactivation was greater in the older, but did not correlate with their behavioural performance. Indeed, older showed similar MI accuracy skills to young participants [39].

Allali et al. [40] have assessed both young and older individuals’ MI skills, using a chronometric test and fMRI. Once again, their results showed increased activation of several brain networks in the older, during MI tasks. Older participants showed greater activation in the right supplementary motor area, the right orbitofrontal cortex, and the left dorsolateral prefrontal cortex. The given instruction focusing on the kinaesthetic modality of MI could contribute to these results, since this modality is associated with more activity in motor-related brain regions. Whereas hyperactivation of frontal areas during MI performance could indicate that older individuals were using executive control mechanisms more than young individuals. All this evidence seems to suggest that older individuals’ greater neural activity could reflect compensation mechanisms for age-related changes in the brain, to preserve their MI skills, as it has been shown for actual execution of movements [41] as well as executive functions and working memory skills [42].

Two other studies confirm this hypothesis [43,44]. It is interesting to note that, during a MI task, Nedelko et al. [43] also found that the activity of ventrolateral premotor cortex and inferior parietal cortex—two seminal regions of the mirror neuron system-, did not change with age. Thus, it seems that mirror neuron activity does not depend on age and provides the basis for novel rehabilitation and therapeutic interventions.

### 4.2. Overcoming fMRI Studies Limitations

Although fMRI studies provide significant new evidence on the evolution of MI during ageing, these studies still show some limitations. Task-based fMRI has been widely adopted to characterize the activity of brain regions involved in movement. However, MI task-based fMRI that includes movement is susceptible to a variety of challenges, such as motion artefacts, which degrade image quality [45]. Moreover, given that MI is a multidimensional process that requires the activity of several brain regions, indeed network studies are represents a necessity [46].

Only two of our selected neuroimaging studies used a different approach compared with fMRI. Neyland et al. [45] used graph theory and network community structured analysis to find differences in MI-related brain connectivity between young and older individuals. Their findings showed that older individuals were characterized by a decreased connectivity in both default mode network (DMN) and sensorimotor network (SMN) during an MI task. DMN decreased connectivity was expected, since this network was originally identified as a collection of regions that show coordinated decreases in activity, during goal-directed and attention-demanding tasks. On the other hand, SMN decreased connectivity was in contrast with previous studies supporting the compensation hypothesis [40]. Lastly, the dorsal attention network (DAN) showed an increased connectivity in elderlies, during MI task. Most researchers agree that the DAN facilitates visuospatial attention and short-term memory. There is also recent evidence that DAN plays a role in spatial orientation. However, these results should be interpreted with caution, since the sample of older people used was composed of over 70-year-olds.

Finally, a study by Burianova et al. [47] used a combination of fMRI and magnetoencephalography (MEG) to assess MI skills in older individuals. The use of MEG allowed them to analyse GABAergic transmission. The results were not consistent with the compensatory hypothesis, showing instead that stronger connectivity to bilateral primary motor cortex in the older group was negatively correlated with their execution performance, but not their imagination one. Overall, an increase in GABA levels was detected. This was due to ageing and could significantly impact the effectiveness of neural communication within the motor system.

The need for more evidence on the evolution of MI during healthy ageing remains a scientific priority. Given the interest of clinicians in MI as a rehabilitation tool, further results are needed to establish whether this skill is indeed preserved during ageing. By combining the different techniques we have analysed so far, we may be able to gain new insights. Table 1 contains a sum up of all the studies we analysed so far.

## 5. Conclusions

The present review is one of the most recent attempts to discuss insights concerning MI age-related changes. Even though MI-based training has therapeutic and rehabilitation potential, research on MI in the older has stalled despite technological and scientific innovations.

We have found that heterogeneity in the experimental protocols, as well as the use of populations with unrepresentative age, is making it challenging to draw unambiguous conclusions about MI skills preservation. Self-report and behavioural task have shown that some MI components are preserved, while others are damaged. Evidence from neuroimaging studies revealed that, during MI tasks, older individual hyperactivate their sensorimotor and attentional networks. Some studies have argued that this represents a compensatory mechanism, others claim that this is a sign of cognitive decline. However, further studies are needed to establish whether MI could be used as a promotion factor to improve cognitive functioning and well-being in older people.

Recently, most scientific study has been focused on pathological older populations, thus neglecting to promote well-being and healthy ageing to prevent physical and cognitive impairments typical of old age.

Several evidence has now recognized physical activity as a crucial environmental factor in cognitive functioning boosting and psychological well-being promotion [4,5,6,48]. It has been proved that an active lifestyle, based on physical activity, could also counteract age-induced cognitive decline [9]. These experimental findings are in line with the clinical panorama, where the concept of “active ageing” is gaining more and more interest. It is known that, with age, the brain’s ability to adapt to the environment decreases gradually, leading to a decline in brain function and neuroplasticity. In a recent study, Rahmi et al. [49] demonstrated that physical activity impacts brain health and cognitive function, as well as cellular and structural changes in older individuals’ brains. Biological changes in the older include increased neurogenesis and synaptogenesis, dendritic remodelling, and synaptic plasticity. They also showed that acute exercise training improves older individuals’ cognitive performance and their quality of life [50].

Enhancing MI skills can replicate the positive effects of physical activity on healthy ageing, since motor imagery and motor execution depend on partially overlapping neural systems [40]. Further, as some evidence on young individuals has shown, physical activity also boosts MI skills [14]. In light of all the methodological issues highlighted in our review, we are still far from concluding that MI-based training is an actually effective option in promoting healthy ageing. However, a greater understanding of how elderlies’ MI work can help us improve movements and gestures that are generally weakened with age.

## Figures and Tables

**Table 1 brainsci-12-01492-t001:** Sum of the studies included in the review. Abbreviation: DMN = default mode network; SMN = sensorimotor network; DAN = dorsal attention network.

MI Components	Studies	Methods	Results
Vividness	Allali et al. 2014 [40]	fMRI/behavioural task	Older showed hyperactivation of motor and executive networks
	Heremans et al. 2011 [23]	Self-report questionnaire	Older and young showed similar vividness skills
	Malouin et al. 2010 [19]	Self-report questionnaire	Older showed lack of vividness skills
	Neyland et al. 2021 [45]	Graph theory and network analysis	Older showed decreased connectivity in DMN and SMN networks; increased connectivity in DAN network
	Saimpont et al. 2015 [17]	Self-report questionnaire	Older and young showed similar vividness skills
	Saimpont et al. 2012 [22]	Self-report questionnaire	Older showed lack of vividness skills
	Schott 2012 [20]	Self-report questionnaire	Older (over 70 years old) showed lack of vividness skills; middle-aged older (between 60 and 69 years old) showed vividness skills comparable to young
Timing	Allali et al. 2014 [40]	fMRI/behavioural task	Older showed hyperactivation of motor and executive networks
	Burianova et al. 2020 [47]	fMRI/MEG/behavioural task	Older showed hyperactivation primary motor cortex and reduce GABA transmission
	Heremans et al. 2012 [26]	Behavioural task	Older and young showed similar timing skills
	Malouin et al. 2010 [19]	Behavioural task	Older and young showed similar vividness skills
	Nedelko et al. 2010 [43]	fMRI/behavioural task	Older showed hyperactivation of sensorimotor networks
	Neyland et al. 2021 [45]	Graph theory and network analysis	Older showed decreased connectivity in DMN and SMN networks; increased connectivity in DAN network
	Saimpont et al. 2015 [17]	Behavioural task	Older and young showed similar timing skills
	Schott 2012 [20]	Behavioural task	Older and young showed similar timing skills
	Zapparoli et al. 2013 [35]	fMRI/behavioural task	Older showed hyperactivation of motor and executive networks
	Zwergal et al. 2012 [44]	fMRI/behavioural task	Older showed hyperactivation of sensorimotor networks
Controllability	Saimpont et al. 2015 [17]	Behavioural task	Older showed lack of controllability skills
	Schott 2012 [20]	Behavioural task	Older showed lack of controllability skills
Accuracy	Devlin & Wilson 2010 [29]	Behavioural task	Older and young showed similar accuracy skills
	Gabbard et al. 2011 [27]	Behavioural task	Older showed lack of accuracy skills
	Saimpont et al. 2010 [28]	Behavioural task	Older showed lack of accuracy skills
	Zapparoli et al. 2016 [39]	fMRI/behavioural task	Older showed hyperactivation of motor and executive networks

## Data Availability

Data are available under request to the corresponding author Laura Mandolesi (laura.mandolesi@unina.it).

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
