# Peer review of "Motor Imagery as a Key Factor for Healthy Ageing: A Review of New Insights and Techniques"

_brainsci, 2022, doi:10.3390/brainsci12111492_

Round 1

Reviewer 1 Report

The manuscript reports on a literature review of studies of age-related changes in motor imagery in elderly adults. Findings are summarized in a relatively clear way, but the description of the review process is lacking in methodological rigor and transparency.

Detailed comments:

In the introduction, the transition from physical activity to MI misses the fact that the health benefits of physical activity are much more than to carry out actions - positive effects to the whole body such as heart, etc.

In the introduction, it is a bit unclear why MI is promoted, why not carry out the action physically? Surely, MI is not more effective than actually carrying out the action?

I miss some rigor and transparency in the method.

What type of review is it – systematic, scoping, or other?

What is the motivation for the restriction in publication year?

What was the search string exactly? Was AND or OR used with the keywords?

Did more than one researcher do the selection? How were conflicts resolved?

What is the rationale for the selected MI components? It is not clear whether this specific list of MI components is the authors’ contribution or taken from the literature.

In the results section, it would be good with an overview of the selected studies at the start.

The conclusion could contain an explicit (short) summary of the findings from the reviewed papers.

Author Response

In the introduction, the transition from physical activity to MI misses the fact that the health benefits of physical activity are much more than to carry out actions - positive effects to the whole body such as heart, etc.      
We thank Reviewer 1 for his/her suggestion. In the Introduction, we have already extensively discussed the beneficial effect of physical activity on older individuals’ wellbeing, and then we have further specified in this revised version why motor imagination can be considered in the same way as physical activity.

In the introduction, it is a bit unclear why MI is promoted, why not carry out the action physically? Surely, MI is not more effective than actually carrying out the action?     
According to Reviewer 1 comment, we have further explained the issue raised. As our reply, we can add that since several evidence reports that execution and imagination of a movement share the same neural networks, the idea that motor imagery (MI), like physical activity, could represent a key factor for counteract cognitive decline associated with ageing, could help to implement MI-based intervention programs. However, this point, wasn’t the focus of our work. We conducted a narrative review to gather information on motor imagery abilities in healthy ageing, examining whether MI and its components are preserved in older individuals.

I miss some rigor and transparency in the method. What type of review is it – systematic, scoping, or other?
Our work is a narrative review. In the revised version of the manuscript, we specified this more clearly in both introduction and method paragraphs.

What is the motivation for the restriction in publication year?      
We found that most of the scientific production on MI was dated within the last decade. Some evidence could be found before our selected timeline, but most of them refers to only behavioural measures and we have chosen to only stick to relatively recent and innovative findings.

What was the search string exactly? Was AND or OR used with the keywords? Did more than one researcher do the selection? How were conflicts resolved?    
Since ours is not a systematic reviews, studies selection and search method were not as rigorous as required by the PRISMA statements. Since MI’s research in healthy ageing still has several open questions, we provided a comprehensive and narrative analysis of MI (in healthy ageing) scientific production to date. Future more systematic studies could be focused on analyzing which methods or MI components is more relevant for healthy ageing.

What is the rationale for the selected MI components? It is not clear whether this specific list of MI components is the authors’ contribution or taken from the literature. 
MI components were selected from the literature. Most of the studies included in our narrative review assessed at least one of these components. In both introduction and 3.1 paragraph we clearly described MI components according to their references. Table 1 also summarize studies that have focused on different components.

In the results section, it would be good with an overview of the selected studies at the start.
Both paragraph 3 and 4 contains our results. Paragraph 3 name was changed due to an oversight in manuscript editing. In both paragraphs we included the number of studies considered, divided thematically (by methods and MI components). The final number of studies selected is reported in the method paragraph.

The conclusion could contain an explicit (short) summary of the findings from the reviewed papers.
In the conclusion paragraph we added a short summary of the results as suggested by Reviewer 1.

Reviewer 2 Report

This manuscript gives a comprehensive review of recent progress during the past decade on how healthy aging changes motor imagery (MI) in human subjects. The authors clearly defined the core concept "MI", and explained the significance of the research of MI on well-beings of elderly people. Importantly, within the manuscript, the authors collected, summarized and commented previous MI studies by the four core components (MI vividness, timing, controllability and accuracy) and three major methodologies (self-report questionnaires, behavior and neuroimaging), through which they explicited the pros and cons by using each of the methods. This review helps give a better understanding of where we are in the present MI research and how MI research is going to benefit well-beings of the elderly people. 

However, there are some minor comments: 

1. Even though the focus of this review was on the role for MI in healthy aging  subjects, it is still important to include studies on populations with cognitive impairment and compare their MI performance to the healthy subject in order to confirm the idea that cognitive decline contributes to impairment of MI. 

Author Response

This manuscript gives a comprehensive review of recent progress during the past decade on how healthy aging changes motor imagery (MI) in human subjects. The authors clearly defined the core concept "MI", and explained the significance of the research of MI on well-beings of elderly people. Importantly, within the manuscript, the authors collected, summarized and commented previous MI studies by the four core components (MI vividness, timing, controllability and accuracy) and three major methodologies (self-report questionnaires, behavior and neuroimaging), through which they explicited the pros and cons by using each of the methods. This review helps give a better understanding of where we are in the present MI research and how MI research is going to benefit well-beings of the elderly people.

However, there are some minor comments:

  1. Even though the focus of this review was on the role for MI in healthy aging subjects, it is still important to include studies on populations with cognitive impairment and compare their MI performance to the healthy subject in order to confirm the idea that cognitive decline contributes to impairment of MI.

We thank Reviewer 2 for their interest and appreciation of our work. Since the focus of our review was on successful and healthy ageing, we decided to avoid including studies on individuals with cognitive impairment, as such an addition would have implied the need to consider numerous other factors (cognitive, environmental and neural) involved in clinically significant cognitive decline. Moreover, we carried out this review from the standpoint of prevention and well-being promotion in old age. The study of MI, as well as of other environmental and cognitive factors, is aimed at providing novel evidence on which can preserve individuals against cognitive impairment associated with ageing. Nevertheless, we do agree with Reviewer 2, and we think that gaining deeper insight into the study of MI in clinical elderly populations is essential, and can certainly be a starting point for any future work.

Reviewer 3 Report

This review looks at the importance of motor imagery abilities for health and wellbeing among older individuals.

I suggest making the title "Motor imagery as a key factor for healthy ageing: an update." "where are we at?" is slang and sounds very unscientific.

Line 21: Are the MI components "damaged" or simply lost through lack of use or practice?

Line 22: Change "individual" to "individuals"

Line 23: Change "represent" to "represents"

Generally "older" is preferred over "elderly"

The Introduction provides a clear and well written background for this review.

Line 110: Delete "only"

Line 124: Change "performs" to "perform"

Lines 128 & 134: By "intermediate" do you mean "middle aged"? 

Line 136 & 142: Here again, I think "older adults" is better than "elderly adults" as you did in lines 144 - 145. This suggestion is relevant throughout the rest of the manuscript.

Line 172: Change "have" to "has"

Line 182: Change "ageing" to "aged"

Line 226: Change "performed" to "perform"

Line 297: Change "MI-related" to "Motor imagery-related"

Line 347: Change "to" to "with"

Line 351: Change "DMN" to Default mode network"

Line 362: Change "MEG" to "magnetoencephalography"

Table 1 is clear and helpful for the reader.

Line 389, 398 & 402: You have spelled "ageing" this way throughout the manuscript. Best to keep it the same.

Author Response

This review looks at the importance of motor imagery abilities for health and wellbeing among older individuals.

I suggest making the title "Motor imagery as a key factor for healthy ageing: an update." "where are we at?" is slang and sounds very unscientific.

We thank Reviewer 3 for his/her suggestion, title has been changed in “Motor imagery as a key factor for healthy ageing: a review of new insights and techniques”.

Line 21: Are the MI components "damaged" or simply lost through lack of use or practice?
In the revised manuscript the term is replaced with “impaired”. In most of our selected studies MI abilities were less efficient in elderly individuals. However, as we discussed in the main text, it’s not easy to draw unambiguous conclusions on the actual preservation of MI during the old age, due to heterogeneous methods and results.

Line 22: Change "individual" to "individuals"        
We modified the term as suggested  

Line 23: Change "represent" to "represents"

We modified the term as suggested  

Generally "older" is preferred over "elderly"

We modified the term as suggested  

The Introduction provides a clear and well written background for this review.

Line 110: Delete "only"

We deleted the term as suggested     

Line 124: Change "performs" to "perform"

We modified the term as suggested  

Lines 128 & 134: By "intermediate" do you mean "middle aged"?

Yes, the term stood for “middle-aged”. We changed the term in the revised manuscript to make it easier to understand.

Line 136 & 142: Here again, I think "older adults" is better than "elderly adults" as you did in lines 144 - 145. This suggestion is relevant throughout the rest of the manuscript.

We modified the term as suggested  

Line 172: Change "have" to "has"

We modified the term as suggested  

Line 182: Change "ageing" to "aged"

We modified the term as suggested  

Line 226: Change "performed" to "perform"

We modified the term as suggested  

Line 297: Change "MI-related" to "Motor imagery-related"

We modified the term as suggested  

Line 347: Change "to" to "with"

We modified the term as suggested  

Line 351: Change "DMN" to Default mode network"

Since we already used the extended term, we prefer to use the abbreviation in this line

Line 362: Change "MEG" to "magnetoencephalography"

We modified the term as suggested

Table 1 is clear and helpful for the reader.

Line 389, 398 & 402: You have spelled "ageing" this way throughout the manuscript. Best to keep it the same.

We modified the term as suggested  

Reviewer 4 Report

Passatello et al. analyzed last decade’s scientific literature on age-related changes in MI skills. It is very interesting work, but as authors mentioned summarizing current status of knowledge is challenging because of the various methodology. authors used conclusive language. I have only two minor comments. It would be useful to add any figure concerning the brain circuits associated with MI revealed in fMRI. What is more, I suggest authors to add short information about the kinds of fMRI and their potential usage in further analysis (Neurol Neurochir Pol 2020;54(1):73-82.) Abbreviations in the Table should be clarified.

Author Response

Passatello et al. analyzed last decade’s scientific literature on age-related changes in MI skills. It is very interesting work, but as authors mentioned summarizing current status of knowledge is challenging because of the various methodology. authors used conclusive language.
We thank Reviewer 4 for his/her appreciation on our work.

I have only two minor comments.     
It would be useful to add any figure concerning the brain circuits associated with MI revealed in fMRI.
We find this suggestion worthy of consideration since it could improve the visual appeal of our work. Unfortunately, it is a time-consuming request, as we would need to gain authors' permission before publishing their images in our review.

What is more, I suggest authors to add short information about the kinds of fMRI and their potential usage in further analysis (Neurol Neurochir Pol 2020;54(1):73-82     

fMRI studies analysed in our review used a combination of different behavioural task and fMRI. We didn’t find any evidence on different fMRI protocol like Diffusion Tensor Imaging (DTI) or other magnetic resonance (MR) protocols. We are sure that innovative MR methods could improve the understanding of MI abilities in older individuals.

Abbreviations in the Table should be clarified.

In the revised manuscript we added clarification on Table 1 abbreviation.

Round 2

Reviewer 4 Report

I have no further remarks.